# Improved LINE-1 Detection through Pattern Matching by Increasing Probe Length

**DOI:** 10.3390/biology13040236

**Published:** 2024-04-02

**Authors:** Juan O. López, Javier L. Quiñones, Emanuel D. Martínez

**Affiliations:** Department of Computer Science, University of Puerto Rico at Arecibo, Arecibo 00612, Puerto Rico; javier.quinones3@upr.edu (J.L.Q.); emanuel.martinez8@upr.edu (E.D.M.)

**Keywords:** LINE-1, ORF, k-mer, probe, pattern matching

## Abstract

**Simple Summary:**

Long Interspersed Element-1 (LINE-1 or L1) is an autonomous transposable element, meaning that its DNA sequences are able to replicate themselves throughout the human genome. This activity may lead to genomic instability and is associated with several different diseases. Additionally, L1s are also capable of replicating other non-autonomous sequences, thereby increasing their disruptive impact. Although there are different tools available that may be used for L1 detection, the heuristics involved affect their accuracy. L1PD (LINE-1 Pattern Detection) uses a novel pattern-matching approach to detect L1s in human genomes, using a fixed set of k-mer probes of length 50 that were generated using the human reference genome GRCh38. This research aims to improve L1PD by using longer probes and testing whether this leads to better results. Additionally, experiments were performed to test the effectiveness of L1PD in detecting L1s in other species, such as dogs, horses, and cows. The results showed that longer probes did improve precision and recall of L1s, not only in humans but in the other species as well.

**Abstract:**

Long Interspersed Element-1 (LINE-1 or L1) is an autonomous transposable element that accounts for 17% of the human genome. Strong correlations between abnormal L1 expression and diseases, particularly cancer, have been documented by numerous studies. L1PD (LINE-1 Pattern Detection) had been previously created to detect L1s by using a fixed pre-determined set of 50-mer probes and a pattern-matching algorithm. L1PD uses a novel seed-and-pattern-match strategy as opposed to the well-known seed-and-extend strategy employed by other tools. This study discusses an improved version of L1PD that shows how increasing the size of the k-mer probes from 50 to 75 or to 100 yields better results, as evidenced by experiments showing higher precision and recall when compared to the 50-mers. The probe-generation process was updated and the corresponding software is now shared so that users may generate probes for other reference genomes (with certain limitations). Additionally, L1PD was applied to other non-human genomes, such as dogs, horses, and cows, to further validate the pattern-matching strategy. The improved version of L1PD proves to be an efficient and promising approach for L1 detection.

## 1. Introduction

### 1.1. Transposable Elements and LINE-1s

Transposable elements (TEs) or Transposons are DNA sequences that move from one location in the genome to another. These elements, occupying about half of the human genome, play an important role in the evolution of genomes, influencing genetic variation and genomic stability. Due to the disruption they cause in the genome, they are linked to various diseases [1,2,3]. In humans, the Long Interspersed Element-1 (LINE-1 or L1) is the only active autonomous TE, accounting for 17% of the genome with more than 500,000 sequences. L1s are capable of mobilizing themselves as well as other non-autonomous TEs, such as Alu and SVA elements, using a “copy-and-paste” mechanism [4,5].

### 1.2. Research Areas Involving LINE-1s

One of the areas where L1s have been of high interest is cancer research, since L1s have been associated with varying forms of cancer [6,7]; in fact, more than 1000 articles focusing on L1s and cancer are available in the PubMed archive [8]. The protein encoded by the Open Reading Frame 1 of L1s (ORF1p), specifically, has been considered a biomarker of neoplasia [9]. ORF1p has also been found to be a candidate biomarker in high-grade serous ovarian carcinoma [10]. In short, ORF1p has shown promise as a multicancer biomarker with potential utility for disease detection and monitoring, including ovarian cancer, gastroesophageal cancer, and colorectal cancer [6,11]. Precisely for this reason, there have been attempts to inhibit ORF1p expression and L1s in general [3,12].

L1s have impacted other research areas besides cancer, including the following recent studies:A study on mice by Song et al. showed that L1-induced hearing impairment could actually be reversed by deleting the L1 retrotransposon insertion [13].A study by Tao et al. showed that L1 insertions can occur frequently at CRISPR/Cas9 editing sites [14].A study by Takahashi et al. suggests L1 activation in the cerebellum may cause ataxia [15].A study by Lou et al. suggests L1s may lead to early spontaneous abortion [16].

Hence, the impact of L1s is wide-reaching, which highlights the importance of further research to better understand how their presence and frequency may be used for disease diagnosis and/or prevention.

### 1.3. L1PD

Because of these adverse effects on health associated with L1s, accurate and efficient L1 detection is important, which is why we developed L1PD (LINE-1 Pattern Detection) [17]. Most L1s are inactive due to rearrangements, point mutations, and truncation [4,5], and that is why L1PD focuses on detecting full-length L1s, which are the most likely to retrotranspose at significant rates. There are several commonly used aligners that use the well-known seed-and-extend strategy, such as BWA-MEM [18], Bowtie2 [19], and CUSHAW2 [20], with certain differences in the seeding and extension techniques [21]. L1PD uses what we have termed seed-and-pattern-match, as opposed to seed-and-extend, where we replaced the heuristics of the extend component with a pattern-matching algorithm. Lopez et al. [17] discuss the shortcomings of seed-and-extend in the context of L1 detection.

The pattern matching is carried out by using a fixed pre-determined set of k-mer probes, generated based on the L1 database L1Base2 [22]. The probes are seeded into a target human genome and then particular patterns of matches are considered to be L1s if they meet certain criteria. Experiments were carried out with varying values for edit distance, distance threshold, and minimum amount of probes required in a pattern. Results were analyzed to determine which values maximized F1 score, and these values were then set to be the default values of L1PD, although the user is able to specify different values if they wish to favor either precision or recall.

Our current research focuses on improving L1PD’s performance by increasing the probe length and automating and improving the probe-generation process, thereby increasing the usefulness of the software and promoting its use in the scientific community to propel further L1 research. Additionally, we tested the effectiveness of L1PD with genomes of other species that are also available in L1Base2, thereby making it possible to detect L1s within other species as long as certain metadata are available.

## 2. Materials and Methods

### 2.1. Probe Generation

The seed-and-pattern-match approach established in L1PD [17] displayed promising results of precision and F1 score when matching the generated 50-mer probes back to the human genome. It was hypothesized that perhaps increasing k-mer length would yield better recall, and this hypothesis was reinforced when it was found that another study by Phan et al. had previously shown a remarkable improvement in recall, with mrFAST and other aligners, when k-mer size was increased from 50 to 75 or 100 [23]. The process of generating the probes will now be explained.

There is a pre-processing step to create a single FASTA file with all of the sequences from which probes are to be generated. First, all of the full-length intact L1s are downloaded from L1Base2 in FASTA format, along with the corresponding Comma-Separated Values (CSV) file containing the metadata for the L1s. The CSV file indicates the positions where the ORF1 and ORF2 reside, allowing us to extract them into two separate FASTA files, one with all of the ORF1s and another with all of the ORF2s. These files are processed separately, producing probes from ORF1 and from ORF2 that will need to be joined (manually) later onto a single file. Additionally, a reference genome is needed (e.g., GRCh38), since it will be used later to weed out probe candidates that do not meet certain criteria. See Figure 1 for a visualization of this pre-processing step.

Once the sequences (ORF1 or ORF2) from which the probes are to be generated are in a single file, the probe generator bash script is executed. First, the sequences provided as input are aligned with the multiple sequence alignment program chosen. The software currently provides support for Clustal Omega 1.2.0 [24], MUSCLE 3.8.31 [25], MAFFT 7.312 [26], and T-Coffee 11 [27]; the user may use whichever they prefer, as long as it is already installed on their system. This was one of the first improvements made to L1PD, since previously the sequences had been manually aligned with the aid of bioSyntax [28]. It should be noted that bioSyntax has continued to be an incredibly useful tool for viewing files on a remote server.

Once the sequences are aligned, the BioPython [29] module’s dumb_consensus function is used to extract k-mers from columns that meet the specified consensus threshold. Note that a high threshold is desirable for better results since this ensures that the probes are indeed good representatives of the L1s they aim to find. Previously, the consensus threshold (sometimes referred to as the “identity percentage” within L1PD files) had been hard coded, but now it is a parameter that may be specified by the user (95% is used by default). Originally it had been difficult to obtain a reasonable amount of probes with 95% identity when using larger values of *k*, but this was no longer an issue once the probe generation was updated to apply the identity percentage as a filtering criterion earlier in the process in order to reduce the number of pre-candidates.

The micro-read Fast Alignment Search Tool (mrFAST) [30] is then invoked to map these k-mers onto the reference genome. The results are analyzed to filter the probe candidates through the following refinement process:Discard k-mers that did not get mapped to the ORF from which they were extracted.From the set of remaining k-mers, discard those that did not map to every single L1.From the set of remaining k-mers, sort in non-descending order of number of map hits on the genome, since a smaller number of hits will lower the number of false positives.Following the order established in the previous step, select as probes the subset of all non-overlapping k-mers.

The resulting FASTA file will include the probes for the corresponding ORF that contained the original sequences. These probes are manually combined with the probes from the other ORF to produce the final probes file. Each comment line in the FASTA file will include some basic information, including the ORF from which it was extracted, the k-mer size, and its offset from the beginning of the ORF1 region. This offset is crucial for the success of L1PD, since it is used to determine whether a pattern of matches corresponds to an L1. Figure 2 provides a visualization of the probe-generation process.

### 2.2. Incorporating Non-Human Genomes

L1Base2 [22] is an online database that contains information for L1s not only for the human reference genome but for the reference genome of several other species as well, such as the *Canis familiaris* (dog) and *Equus caballus* (horse). Besides improving the performance of human genomes, one of the main goals for this updated version of L1PD was to be able to generate probes and detect L1s for these other species as well.

The code structure for the probe-generation process allows probes for other species to be generated without any additional effort. Once the appropriate files (FASTA and CSV) are downloaded from L1Base2, the process works seamlessly to extract the L1 components and then generate the probes. However, it should be noted that the amount of probes, and the time required to generate them, may vary considerably.

### 2.3. L1PD Algorithm

The L1PD algorithm [17] receives a fully-assembled genome and uses mrFAST [30] to index the genome and then map the probes against that genome, generating a SAM file. It should be noted that most of the run time of L1PD is taken by these steps. The SAM file, along with the probes, are then fed into a Python script (L1PD.py), which applies the pattern-matching algorithm and generates an output file in GFF3 (General Feature Format Version 3) format. To summarize, the pattern-matching algorithm consists of finding patterns of “hits” of probes (in the SAM file) that are in the same order and within the expected distance as specified by the FASTA file that contains the probes. The full algorithm is explained by López et al. [17], but Figure 3 provides an overall visualization of L1PD. Note that the edit distance, distance threshold, and min. amount of k-mers in a pattern are all parameters that may be specified by the user, although L1PD provides default values for certain species.

GFF3 files store genome information features in nine tab-delimited columns. Of the nine columns, L1PD fills the following seven:sequence id (chromosome where LINE-1 was found);source (“L1PD”);type (“mobile_genetic_element”);start (start position of the LINE-1);end (end position of the LINE-1);strand (“+” for forward strand and “−” for reverse strand);attributes (“Name = LINE1”).

One of the main obstacles in extending L1PD for use with other species was the fact that not only do different species have different amounts of L1s, but the average L1 length also varies, as well as the average lengths of the main components (ORFs and UTRs). These lengths are important because they are used to calculate the start and end positions that will be written to the GFF3 file. Since the lengths are expected to be normally distributed, the updated version of L1PD now uses the CSV file to calculate the modes (most common value) of the length of each component, which are then used to calculate the start and end positions of the L1.

Since the previous version of L1PD focused only on human genomes, it also generated a histogram comparing the number of L1s in each chromosome of the GRCh38 vs. the provided target genome. However, this feature has been removed since now the user will be able to run L1PD with different species.

### 2.4. Precision and Recall

The precision and recall shell script was initially used as a private component but is now also being provided in the repository for public use. This component was created to confirm the validity of L1PD and also to test its sensitivity to changes in edit distance, distance threshold, and min. amount of probes required in a pattern. The aim was to apply L1PD to the reference genome with varying combinations of these parameters, calculating the precision, recall, and F1 score each time, and then use the values that maximized F1 score as default values for L1PD. It should be noted that the F1 score is the harmonic mean of precision and recall [31]. Additionally, tables with detailed results were provided [17] to help guide the user as to what parameters to change in case they wished to favor either precision or recall.

Another of the improvements to L1PD was changing the primary parameters (distance, threshold, and min. amount of probes) from being fixed to allowing the user to specify them on the command line, which is necessary since these values are expected to vary considerably with genomes of different species. This provides the user the flexibility to execute the code and run hundreds of experiments consecutively, exploring the results for different combinations of these parameters. These experiments are essential to understanding how well the probes are performing for that particular species.

Since L1Base2 [22] contains all of the relevant L1 information, it was used to calculate the total amount of L1s in humans, thereby allowing us to calculate the amount of true positives and false negatives in order to properly calculate the values for precision, recall, and F1 score. However, these values were fixed, so L1PD was updated so that the user may provide the directory where the corresponding L1Base2 CSV files are stored; that way, these calculations may still be performed for different species. Through testing, it was found that there were a few entries in the CSV files for the human genome that were empty. The code now gracefully disregards these special cases to allow the process to continue. Note that the CSV files must follow a particular naming convention; sample files will be provided in the repository.

## 3. Results

### 3.1. Results with Different k-mer Sizes

The algorithms were executed on the human genome using three different k-mer sizes (50, 75, 100) in order to assess the best outcome in F1 scores. For these experiments, the identity percentage stayed at 95%, as it displayed promising results, although this parameter can be adjusted when executing the algorithm. To test the effectiveness of the probes with the updated probe-generation algorithm, L1PD was executed using varying ranges of edit distance, distance threshold, and min. amount of k-mers in a pattern. For the 50-mer probes, the best F1 score was found with an edit distance of 15, whereas the 75-mer and 100-mer probes both peaked at an edit distance of 30. Table 1 summarizes these results; detailed results are available in Section A.1.

As can be seen, the 100-mers obtained the highest F1 scores. Both 75-mers and 100-mers showed relatively similar values once the edit distance reached at least 15, but 100-mers have the fewest probes of the three different k-mer sizes and require fewer k-mers per pattern, which was expected due to the increase in k-mer size. The overall highest case of F1 score was achieved with the 100-mers using edit distance 30, threshold 600, and a minimum of nine probes. For all three k-mer sizes, the results peaked and then started steadily decreasing. Figure 4 provides a visual comparison of the best F1 score for each edit distance and k-mer size.

Additionally, there was no change in time complexity when using larger probes, as shown in Table 2.

### 3.2. L1PD with Other L1Base2 Genomes

Given that L1PD is based on L1Base2 [22], its applicability extends to various genomes. We chose to apply it to species from different orders: Carnivora, represented by the dog (*Canis Familiaris*); Artiodactyla, represented by the cow (*Bos taurus*); and Perissodactyls, represented by the horse (*Equus caballus*). The reference genomes for these species were obtained from the Ensembl Release 84 FTP site (http://ftp.ensembl.org/pub/release-84/fasta/ (accessed on 31 December 2023), ensuring that the genomes used are the same versions as those employed by L1Base2. Additionally, other files, such as the L1 sequences and metadata files, were obtained from L1Base2 in the same manner as with the human genome.

Using the same process as for the human genome, probes were generated for each species, with three different k-mer sizes (50, 75, and 100) and using a consensus threshold of 95%. Once the probes had been generated, L1PD was then executed with no changes to the code, but this time the arguments were the corresponding files for the species being targeted. The precision and recall experiments were then carried out to determine the values of k-mer size, edit distance, distance threshold, and minimum amount of probes per pattern that maximized the F1 score. The summarized results for dog and horse are presented in Table 3 and Table 4, respectively. Tables with more detailed results can be found in the appendix (Section A.2 and Section A.3). Additionally, Figure 5 and Figure 6 provide a visual comparison of the best F1 Score for each edit distance and k-mer size for the dog and cow genomes, respectively.

Although good results were obtained with the cow genome using 50-mers and an identity percentage of 95%, probes for ORF1 could not be generated with a k-mer size of 75. If we consider the other species (human, dog, and horse), it is a common trend for the total number of probes to reduce as the k-mer size increases. However, there were only five 50-mer probes for the cow, so perhaps to obtain 75-mer probes, the identity percentage might need to be reduced. Due to this setback and pending further analysis, L1PD was executed exclusively with probes of size 50 for the cow. Nonetheless, the results for the cow were the most promising, yielding the highest F1 score of all when using 50-mers (0.66801). Summarized data for the cow are presented in Table 5; more detailed results can be found in the appendix (Section A.4).

As with the human genome, the time required to execute L1PD was not affected significantly by using larger probes, as shown in Table 6.

## 4. Discussion

The work realized by Phan et al. [23] established that the probe k-mer size and recall score have a positive correlation, meaning that increasing the k-mer size should result in an increase in recall. The initial 50-mer experiments were thus adjusted to include 75-mers and 100-mers following the hypothesis that increasing the probe length will directly improve the pattern-matching recall. The results support the hypothesis previously established, although the difference was not as significant as expected. Once adequate values of edit distance, distance threshold, and minimum probes were met, the recall gradually increased for larger k-mer sizes. In addition, the precision increased as well (when compared to 50-mers), resulting in an overall higher F1 score. The overall best results with humans were found with 100-mers, using an edit distance of 30, a distance threshold of 600, and a minimum of 9 probes per pattern, for an F1 score of 0.72931.

Increasing edit distance provides flexibility in finding probes that may have had more changes due to rearrangements or mutations, which is why the lowest edit distance always provided the worst results. Similarly, a higher distance threshold allows for probe hits that may be further (or closer) away than originally expected, due to insertions (or deletions) that may have occurred. As anticipated, increasing recall can have the undesirable side effect of decreasing precision, and this can be seen in certain cases when comparing results from 100-mers with results of 75-mers.

On the other hand, it follows that increasing k-mer size would provide better results. DNA contains highly repetitive sequences, so a larger k-mer size can reduce how often these common sequences will be matched and more accurate results can be obtained.

### 4.1. Advantages and Limitations

#### 4.1.1. Advantages

One of the main reasons L1PD was originally developed was because the seed-and-extend strategy seemed ill-fitted for the task of L1 detection [17]. L1PD removes the heuristics associated with the extended phase and replaces it with pattern matching, making it a novel and promising approach for L1 detection.

The probe-generation scripts can be executed with other genomes annotated in L1Base2, allowing researchers to generate probes for other genomes. These probes, along with the corresponding CSV files, can then be used by L1PD to detect L1s.

Because of the narrow focus of L1PD, users are able to execute it in “Genome mode” by providing very little input. By default, L1PD uses a previously generated set of probes and the values of edit distance, distance threshold, and minimum amount of k-mers that resulted in optimal F1 scores (see Appendix A for the experimental results). Hence, the user only needs to provide the genome (in FASTA format) in which L1s are to be detected and to specify the species of that genome. L1PD also allows for the user to specify custom values of any of these parameters, in case they wish to favor either precision or recall.

#### 4.1.2. Duplicate Matches

Through experimentation, it was found that excessively high values of edit distance and/or distance threshold generated “duplicate” matches, meaning that certain patterns were matched to more than one L1. However, this did not affect the results since the highest F1 score was obtained with lower edit distance and threshold values that did not exhibit this behavior. In the future, additional experiments could be carried out to get a better understanding of the values of the parameters that cause these duplicates to start creeping in. This might help to limit upper boundaries for edit distance and/or threshold since currently the upper bounds are determined experimentally. For example, although for 50-mers we carried the experiments through a maximum edit distance of at least 25, this might be too high since it could mean that half of the k-mer has been changed.

#### 4.1.3. CSV File Required

The original version of L1PD was designed specifically for humans, so there was certain information that was hard coded into the software, such as the average ORF1 and ORF2 lengths. This information can no longer be hard coded since L1PD can now be easily used with different genomes available from L1Base2. Due to this, L1PD now requires an additional argument with the path to a CSV file containing the metadata for the reference genome of that species. These files must follow the format of the CSV files available from L1Base2, and L1PD will use that information to determine the mean length of the different L1 components.

The long-term goal is for researchers to be able to use L1PD with other species they are interested in. However, this will require the CSV file with the metadata to be created, which is a limitation in applying L1PD to different species. One possible workaround is to avoid GFF3 format and/or use an annotation format that does not require the start and end position of every L1.

#### 4.1.4. Time Required for Generating Probes

When we began applying L1PD to other species from L1Base2, the intention was to execute it with at least one representative from each order available on the same webpage. For rodents, the mouse (*Mus musculus*) was chosen as the representative. However, when attempting to execute the steps for generating ORF2 probes with a k-mer size of 50, the process had already been running for more than 20 h, significantly longer than all the previous species. We believe this extended execution time may be due to the substantial number of full-length intact L1s in the mouse, totaling 2811; much higher than the second-highest species we worked with, the dog, which only had 264 full-length intact L1s. Upon surpassing the 20-h mark, we decided to stop the execution of the script, setting a future goal of optimizing the code used in probe generation. Besides analyzing the code structure, one of the ideas for optimization we implemented is to allow the user to specify the number of threads used by the alignment program, provided that the program supports such an option. However, mrFAST [30] currently does not support multithreading.

#### 4.1.5. Finding Appropriate Values of *k*

Finally, as seen with the cow genome, 100-mers are not always feasible. On the other hand, it is possible that certain genomes could use longer k-mers, so it requires a bit of experimentation for the user to determine what size k-mer to use. Work has been started on polishing the probe-generator component so that it may be used to find optimal values of *k*, thereby saving time and promoting the use of adequately sized k-mers.

### 4.2. Applications

As with the original version [17], the updated version of L1PD can be used to calculate Copy Number Variation by analyzing the change in L1 count with respect to the reference genome.

Since L1PD can now be applied to several genomes, the histogram feature has been temporarily disabled, but perhaps it can be enabled in the future to get a visual comparison of the L1 distribution in the user-provided genome in comparison to the reference genome. Nonetheless, L1PD can still be executed in three different modes, allowing its use even when only reads in FASTQ format are available. See Figure 7 and Lopez et al. [17] for more details.

The pattern-matching strategy lends itself to be applied to different families of sequences that have column ranges of high similarity. This is the reason why the methodology was thoroughly explained and the code is made freely available. We plan on expanding in this area in the near future and possibly collaborating with colleagues who work with different species and/or families of sequences.

### 4.3. Future Work

Future work includes optimizing the code, finding the largest possible k-mer size for humans that yields the best results in terms of maximizing recall, trying to apply the pattern-matching algorithm to other families of sequences, such as SINEs (Short Interspersed Nuclear Elements), trying to make L1PD easier to use with genomes not available from L1Base2, and polishing the probe generator component to be used as a standalone tool.

## 5. Conclusions

L1PD proves to be an efficient and promising approach for L1 detection through its seed-and-pattern-match approach. Increasing k-mer size from 50 to 100 improved precision and recall, and L1PD works adequately, although with lower-than-expected results, with genomes from other species available from L1Base2. By improving L1PD’s performance and usefulness, as well as that of the probe-generation algorithm, we hope to help propel further L1 research.

## Figures and Tables

**Figure 1 biology-13-00236-f001:**
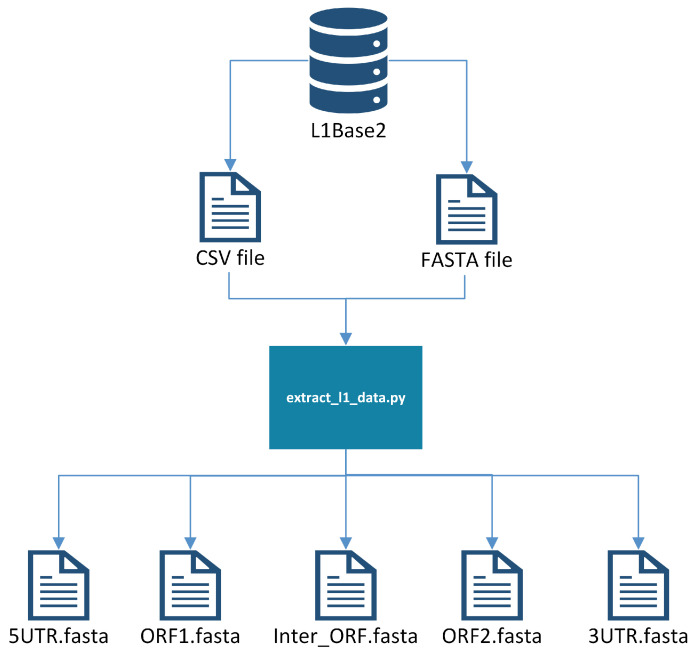
Extracting L1 components into separate files.

**Figure 2 biology-13-00236-f002:**
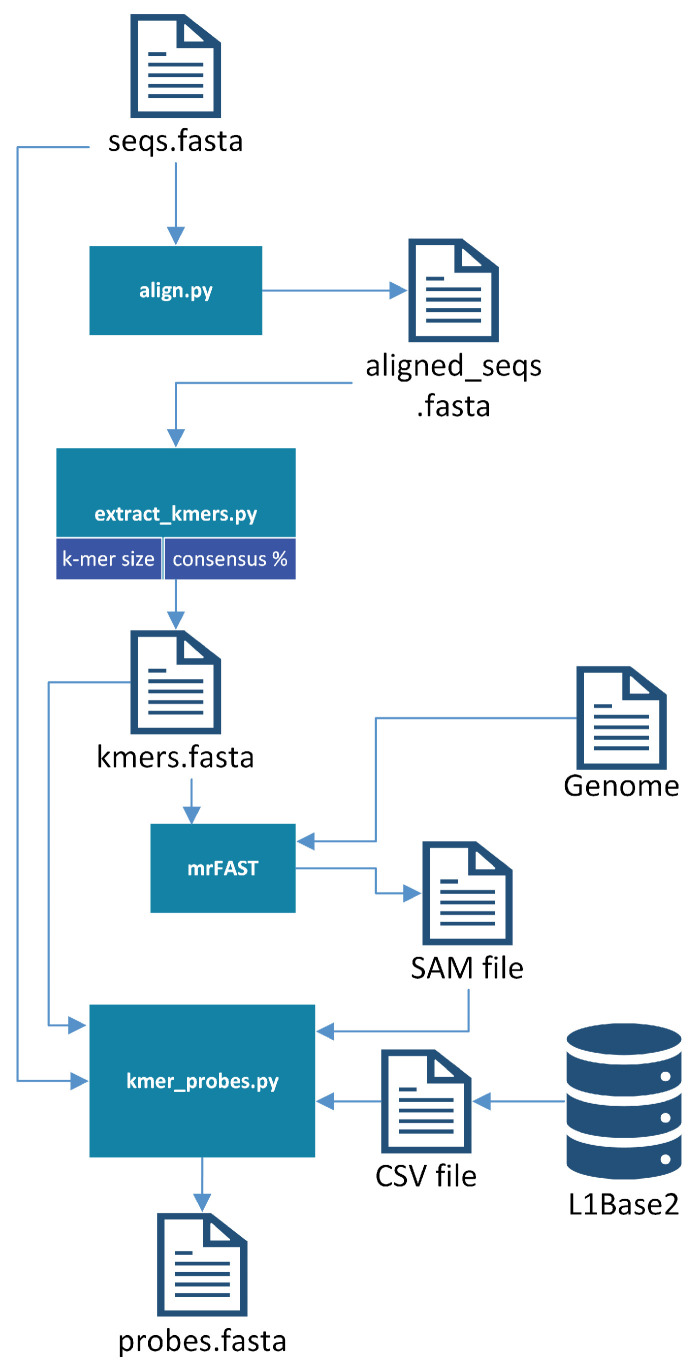
Probe-generation process.

**Figure 3 biology-13-00236-f003:**
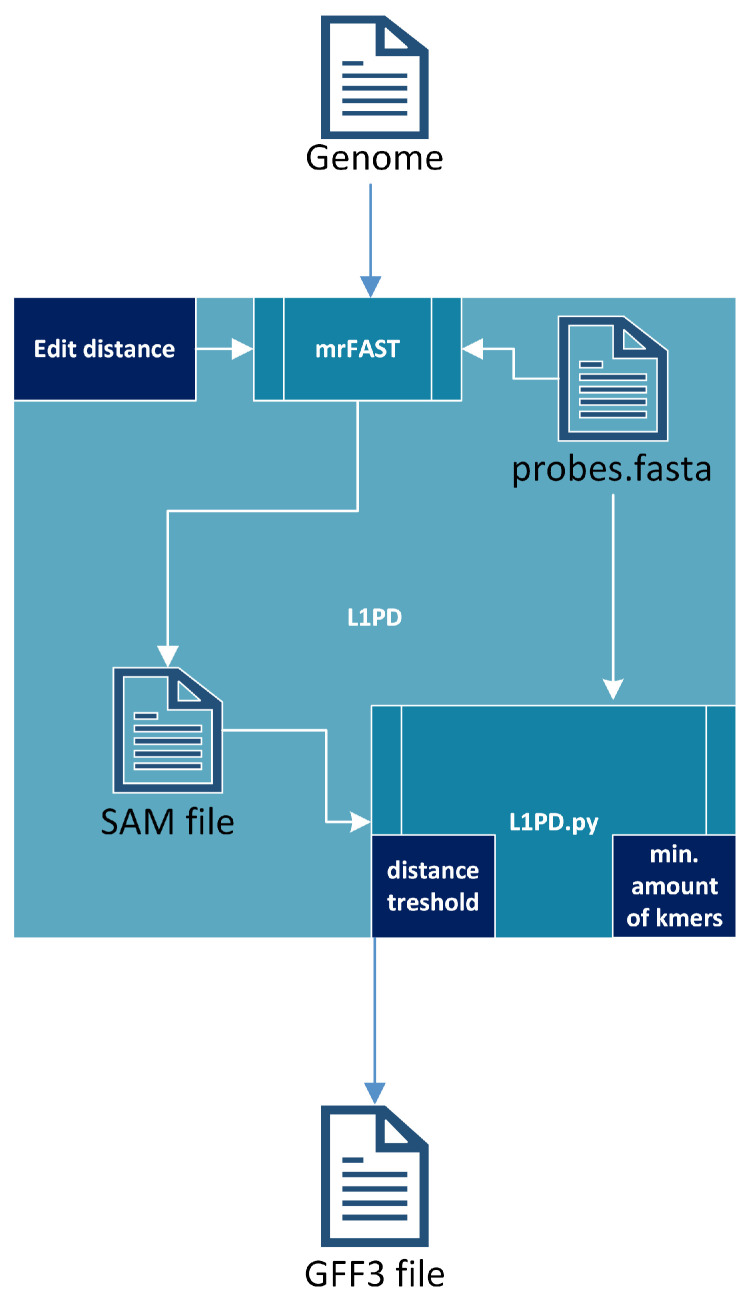
L1PD algorithm.

**Figure 4 biology-13-00236-f004:**
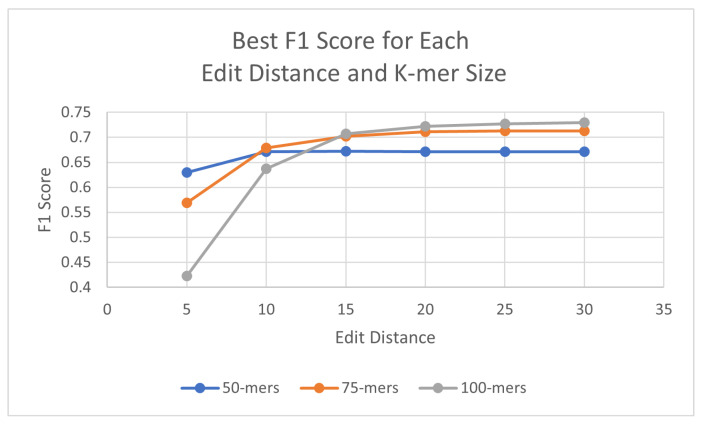
Best F1 scores for each edit distance and k-mer size in human genome.

**Figure 5 biology-13-00236-f005:**
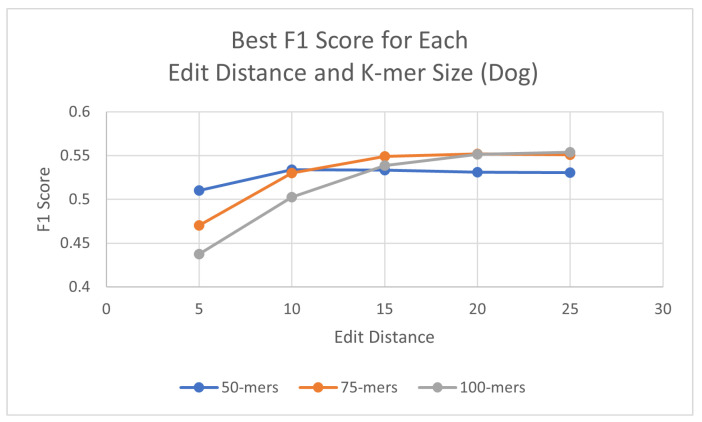
Best F1 scores for each edit distance and k-mer size in dog genome.

**Figure 6 biology-13-00236-f006:**
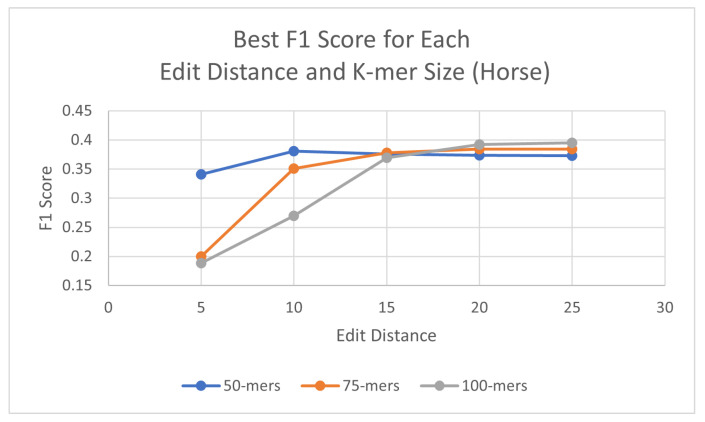
Best F1 scores for each edit distance and k-mer size in horse genome.

**Figure 7 biology-13-00236-f007:**
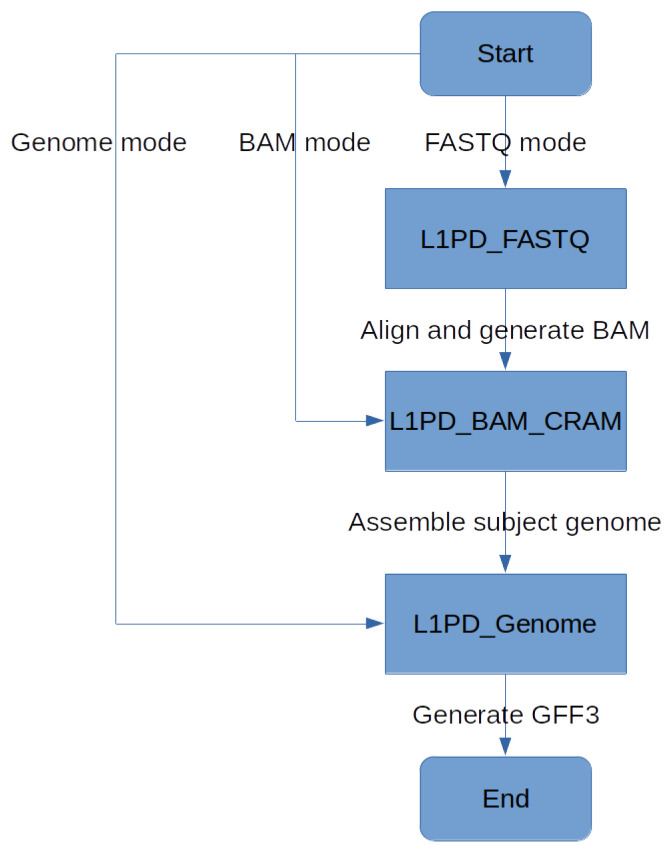
L1PD mode flowchart.

**Table 1 biology-13-00236-t001:** Best F1 score by edit distance and k-mer size: *Homo sapiens* (human).

Edit Distance	k-mer Size	Threshold	Minimum Probes	Precision	Recall	F1 Score
	**50**	**675**	**24**	**0.79044**	**0.50574**	**0.61682**
5	75	700	12	0.80344	0.44042	0.56895
	100	575	9	0.95448	0.27152	0.42277
	50	650	24	0.76713	0.59278	0.66877
10	**75**	**625**	**14**	**0.79380**	**0.59249**	**0.67852**
	100	625	7	0.76994	0.54348	0.63718
	50	650	24	0.76265	0.59842	0.67062
15	75	625	17	0.89338	0.57801	0.70189
	**100**	**625**	**9**	**0.89072**	**0.58613**	**0.70701**
	50	675	25	0.77723	0.58876	0.66999
20	75	625	17	0.88651	0.59366	0.71111
	**100**	**625**	**9**	**0.88124**	**0.61173**	**0.72215**
	50	675	25	0.77637	0.58942	0.67009
25	75	625	17	0.88086	0.59871	0.71287
	**100**	**600**	**9**	**0.87538**	**0.62226**	**0.72742**
	50	675	25	0.77682	0.58942	0.67026
30	75	625	17	0.87747	0.60083	0.71326
	**100**	**600**	**9**	**0.86826**	**0.62870**	**0.72931**

Values in bold indicate the parameters that resulted in the best F1 Score for each edit distance. The row highlighted in cyan represents the highest F1 Score obtained across the entire species.

**Table 2 biology-13-00236-t002:** Time required by L1PD with different k-mer sizes with default values: *Homo sapiens* (human).

k-mer Size	Edit Distance	Distance Threshold	Minimum k-mers Required	Total Amount of k-mers	Time
50	15	625	18	34	44 min, 33 s
75	30	625	17	22	44 min, 28 s
100	30	600	9	13	44 min, 29 s

**Table 3 biology-13-00236-t003:** Best F1 score by edit distance and k-mer size: *Canis Familiaris* (dog).

Edit Distance	k-mer Size	Threshold	Minimum Probes	Precision	Recall	F1 score
	**50**	**750**	**5**	**0.58541**	**0.45217**	**0.51022**
5	75	900	2	0.57946	0.39582	0.47038
	100	825	2	0.65690	0.32825	0.43775
	**50**	**725**	**10**	**0.65220**	**0.45217**	**0.53406**
10	75	750	6	0.65151	0.44686	0.53011
	100	825	2	0.58119	0.44245	0.50241
	50	750	10	0.64183	0.45618	0.53331
15	**75**	**725**	**9**	**0.71589**	**0.44565**	**0.54933**
	100	750	6	0.68649	0.44325	0.53868
	50	750	9	0.63331	0.45748	0.53122
20	**75**	**725**	**9**	**0.70552**	**0.45327**	**0.55193**
	100	725	7	0.70385	0.45347	0.55156
	50	725	11	0.65070	0.44826	0.53082
**25**	75	725	9	0.69685	0.45588	0.55116
	**100**	**725**	**7**	**0.69503**	**0.46019**	**0.55373**

Values in bold indicate the parameters that resulted in the best F1 Score for each edit distance. The row highlighted in cyan represents the highest F1 Score obtained across the entire species.

**Table 4 biology-13-00236-t004:** Best F1 score by edit distance and k-mer size: *Equus caballus* (horse).

Edit Distance	k-mer Size	Threshold	Minimum Probes	Precision	Recall	F1 Score
	**50**	**1375**	**2**	**0.32847**	**0.35432**	**0.34089**
5	75	1575	2	0.43180	0.13032	0.20020
	100	1575	2	0.48668	0.11694	0.18856
	**50**	**1375**	**7**	**0.35165**	**0.41498**	**0.38096**
10	75	1525	2	0.32324	0.38356	0.35082
	100	1575	2	0.45382	0.19156	0.26939
	50	1650	8	0.34442	0.41396	0.37599
15	75	1550	5	0.36774	0.38909	0.37810
	**100**	**1425**	**2**	**0.36161**	**0.37818**	**0.36969**
	50	1650	9	0.36001	0.38865	0.37377
20	75	1550	6	0.38220	0.38603	0.38410
	**100**	**1475**	**4**	**0.38428**	**0.40116**	**0.39253**
	50	1650	9	0.35720	0.39025	0.37298
**25**	75	1550	6	0.37559	0.39345	0.38431
	**100**	**1350**	**5**	**0.39556**	**0.39447**	**0.39501**

Values in bold indicate the parameters that resulted in the best F1 Score for each edit distance. The row highlighted in cyan represents the highest F1 Score obtained across the entire species.

**Table 5 biology-13-00236-t005:** Best F1 score by edit distance for k-mer size 50: *Bos taurus* (cow).

Edit Distance	k-mer Size	Threshold	Minimum Probes	Precision	Recall	F1 Score
5	50	900	4	0.88203	0.48694	0.62747
10	50	900	4	0.86544	0.54644	0.66990
15	50	900	4	0.86354	0.56066	0.67989
20	50	900	4	0.85413	0.56330	0.67887
**25**	**50**	**900**	**4**	**0.86045**	**0.56462**	**0.68182**

The row highlighted in cyan represents the highest F1 Score obtained across the entire species.

**Table 6 biology-13-00236-t006:** Time required by L1PD with different k-mer sizes in different genomes with default values.

Species	k-mer Size	Number of k-mers	Time Used by L1PD
	50	24	38 min, 8 s
Dog	75	14	48 min, 24 s
	100	10	44 min, 4 s
	50	32	49 min, 27 s
Horse	75	13	52 min, 25 s
	100	9	48 min, 32 s
Cow	50	4	52 min, 24 s

## Data Availability

The source code for L1PD, as well for the probe generation and precision and recall processes, is available at https://github.com/juan-lopez/L1PD (accessed on 31 December 2023). The code consists of several shell scripts, Python scripts, FASTA files with the probes, as well as sample output files. The shell scripts should run under most Unix-like systems. L1PD may be executed in one of three modes: 1. Genome mode; 2. BAM/CRAM mode; 3. FASTQ mode. BAM/CRAM mode automatically invokes Genome mode, and FASTQ mode automatically invokes BAM/CRAM mode, as shown in Figure 7.

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
