# Peer review of "Improved LINE-1 Detection through Pattern Matching by Increasing Probe Length"

_biology, 2024, doi:10.3390/biology13040236_

Round 1
Reviewer 1 Report
Comments and Suggestions for Authors
Overall, I find the manuscript well-written and the improved version of L1PD appears to be a promising approach for LINE-1 detection. However, to enhance the clarity and completeness of the manuscript, I suggest the authors discuss the advantages and limitations of their algorithm in more detail. Providing insights into the strengths and potential applications, as well as any constraints or scenarios where the algorithm may be less effective, would contribute to a more thorough understanding for the readers. This additional information could further strengthen the manuscript and assist the readers in assessing the algorithm's utility in different contexts. Please consider incorporating a brief discussion on the advantages and limitations of the proposed method.
Comments on the Quality of English LanguageMinor editing of English language required
Author Response
Per the suggestion, the Discussion section was restructured to include an Advantages and Limitations subsection where we elaborated on this topic. Additionally, a brief Applications subsection was also added.
Reviewer 2 Report
Comments and Suggestions for Authors
The manuscript by Lopez et al. describes the refinement of a bioinformatic tool, L1PD, to identify L1 Long Interspersed Element (LINE-1) repeats in mammalian genomes. The authors indicate the improvements to the algorithm that they tested (i.e. increasing k-mer size of the probes, creating a novel module to generate the probes from database hits, etc) and their effect on precision and recall.
The article is well written. My onsly suggestion is that Tables 1-3 are provided as supplementary data, preserving only Table 4 in the main text.
Author Response
Per the suggestion, Tables 1-3 were moved to Appendix A.
Reviewer 3 Report
Comments and Suggestions for Authors
The manuscript by López et al describes some improvements of a previously described approach for detection of LINE-1 elements through pattern matching. Better results are evidenced by higher precision and recall when the probe length is increased from 50 to 75 or to 100. An additional improvements regards the possibility to probe additional non-human genomes. According to the authors, the better detection is helpful for investigating the roles of L1 in diseases.
The paper is clear and well written. I am not sure if the improvement is that significant and if all the large tables are required but the paper could be published in Biology. I was a bit surprised by the sentence "On a few rare occasions this generated results that were not feasible, such as obtaining a precision of more than 100%." (p13 l261). Could this problem not be corrected or better explained?
Author Response
Several of the large tables were moved to Appendix A.
The sentence referring to the “more than 100%” precision was removed and it was clarified that these duplicate matches did not affect the results, since the highest F1 Score was always obtained with lower edit distance and threshold values that did not exhibit this behavior.
A brief comment in the Discussion section was added indicating that the difference in recall was not as significant as expected, and in Future Work we added that we will continue working to find a higher k-mer size to get the best recall results possible.
Reviewer 4 Report
Comments and Suggestions for Authors
The authors provide an improved version of L1PD method. Results show that the performance of the L1PD algorithm is improved by increasing the k-mer probe length. However, the overall quality of the paper is poor. The paper is not well-written and organized. The novelty of the work is very limited. And not enough experiments were performed to provide a comprehensive view of the algorithm.
Major
1. The introduction is not well organized to provide a general view of current studies in the files. Listing four papers doesn't highlight the importance of further research. And section 1.3 mainly describes the work already published in other journal, which is not appropriate to be included here, taking such a large portion.
2. The method section needs to be written. The description is very wordy. Current version is more like a technique report for the author self instead of an academic paper.
3. The major contribution of the work is replacing part steps used in L1PD with automatic tools and incorporating data other than human. The novelty is very limited, which is far from being considered for publication.
4. The details of the method part are unclear. For example, in line 112, k-mers that have the smallest number of map hit were selected as probe. What is the smallest number of hit? What if the smallest hit is very high? Is a threshold applied here?
5. In line 104, why use 95% (default value) as a filtering criteria, then the previous issue that identifying a reasonable amount of probes with 95% to identify is difficult is not an issue again.
6. It can be expected that increasing the k-mer length will improve the performance. But the results shows limited improvement with F1 score from about 0.67 to 0.71 by increasing the k-mer length from 50 to 100. The work is lack of significance for the field.
7. By increasing the probe length, how will the time complexity change?
8. The author expanded the database to dog and horse. However, mice or monkey are more often used in the studies. The expansion doesn't show much benefit to the field.
Comments on the Quality of English LanguageThe overall structure of the paper is poor. The paper is not written formally and academically.
Author Response
-
The introduction is not well organized to provide a general view of current studies in the files. Listing four papers doesn't highlight the importance of further research. And section 1.3 mainly describes the work already published in other journal, which is not appropriate to be included here, taking such a large portion.
Response:
The four papers that you mention are merely examples of non-cancer research areas, but in the paragraph that precedes those references we state (and cite the reference) that there are more than 1,000 articles focusing on L1s and cancer.
Because the purpose of this article is to discuss improvements of L1PD, section 1.3 introduces what L1PD is. Section 1.3 only takes half of a page, which we feel is adequate. -
The method section needs to be written. The description is very wordy. Current version is more like a technique report for the author self instead of an academic paper.
Response:
One of the aims of the paper is to explain the process of generating the probes, and this is the part that is wordy. However, two images were provided to help get the point across. -
The major contribution of the work is replacing part steps used in L1PD with automatic tools and incorporating data other than human. The novelty is very limited, which is far from being considered for publication.
Response:
Part of the novelty is indeed automating part of the process and incorporating non-human genomes. Additionally, L1PD was improved to use larger k-mers, improving precision and recall, improving the probe generation process, and to thoroughly discuss the process to perhaps inspire others to see whether a pattern-matching approach might be of interest. -
The details of the method part are unclear. For example, in line 112, k-mers that have the smallest number of map hit were selected as probe. What is the smallest number of hit? What if the smallest hit is very high? Is a threshold applied here?
Response:
This section was rephrased to try to make it clearer. -
In line 104, why use 95% (default value) as a filtering criteria, then the previous issue that identifying a reasonable amount of probes with 95% to identify is difficult is not an issue again.
Response:
This section was rephrased to try to make it clearer. -
It can be expected that increasing the k-mer length will improve the performance. But the results shows limited improvement with F1 score from about 0.67 to 0.71 by increasing the k-mer length from 50 to 100. The work is lack of significance for the field.
Response:
In the Discussion section it was added that the change was not as significant as had been expected, and in the Future Work section it mentions that we will continue working to find a way to get better recall results. -
By increasing the probe length, how will the time complexity change?
Response:
Per the suggestion, tables were added to sections 3.1 and 3.2 showing there was no change in time complexity. -
The author expanded the database to dog and horse. However, mice or monkey are more often used in the studies. The expansion doesn't show much benefit to the field.
Response:
The article specifically indicates the difficulty we had with the mouse genome, and that text has been moved to the new Limitations subsection of Discussion. By thoroughly explaining the process and making the code available, it is relatively simple for a research to apply L1PD to the monkey genome if they wish to do so. That is precisely our goal, for researchers to use L1PD in other genomes and for our pattern-matching strategy to promote further research.
Round 2
Reviewer 4 Report
Comments and Suggestions for Authors
The primary deficiency of the study lies in its absence of novelty. Although the study does make contributions such as expanding the k-mer size, substituting certain L1PD steps with automated tools, integrating non-human data, and providing discussions that may inspire new ideas in the field, these do not sufficiently distinguish it from the one that has published the L1PD algorithm. The focus of the current study on increasing the k-mer parameter for performance enhancement provide limited contribution and interest to the field. Unless substantial modifications are implemented, the current work does not meet the criteria for publication.
Author Response
We feel that, although the results and ideas presented in this update to L1PD might not be groundbreaking, they serve as a valid contribution to the scientific community. In fact, a couple of months ago we received an e-mail from a professor in the United States who has a lab specializing in transposable elements. He wrote “Thank you for developing L1PD! I found your method from the TEhub. I tried it out in the human genome and it performed very nicely! Thank you for making a tool that actually runs the first time I tried it.” We wrote back indicating the most recent work, including the increase in k-mer size and the integration of non-human genomes, and his response included statements such as “I am excited to learn that you are working on extending the functionality of L1PD” and “So it’s exciting to learn that you are including more genomes in the L1Base2”. This is certainly a particular example, but we feel it serves as motivation that our work can indeed propel further research that could have more impact in the scientific community. We look forward to continuing improving L1PD and other related tools to help other scientists pursue additional research.